# Advancing Human Motion Recognition with SkeletonCLIP++: Weighted Video Feature Integration and Enhanced Contrastive Sample Discrimination

**DOI:** 10.3390/s24041189

**Published:** 2024-02-11

**Authors:** Lin Yuan, Zhen He, Qiang Wang, Leiyang Xu

**Affiliations:** Department of Control Science and Engineering, Harbin Institute of Technology, Harbin 150001, China; _eunseo_v@hit.edu.cn (L.Y.); hezhen@hit.edu.cn (Z.H.); xuleiyang@stu.hit.edu.cn (L.X.)

**Keywords:** action recognition, multi-modal fusion, contrastive learning

## Abstract

This paper introduces ‘SkeletonCLIP++’, an extension of our prior work in human action recognition, emphasizing the use of semantic information beyond traditional label-based methods. The innovation, ‘Weighted Frame Integration’ (WFI), shifts video feature computation from averaging to a weighted frame approach, enabling a more nuanced representation of human movements in line with semantic relevance. Another key development, ‘Contrastive Sample Identification’ (CSI), introduces a novel discriminative task within the model. This task involves identifying the most similar negative sample among positive ones, enhancing the model’s ability to distinguish between closely related actions. Incorporating the ‘BERT Text Encoder Integration’ (BTEI) leverages the pre-trained BERT model as our text encoder to refine the model’s performance. Empirical evaluations on HMDB-51, UCF-101, and NTU RGB+D 60 datasets illustrate positive improvements, especially in smaller datasets. ‘SkeletonCLIP++’ thus offers a refined approach to human action recognition, ensuring semantic integrity and detailed differentiation in video data analysis.

## 1. Introduction

In the field of human action recognition, deep learning techniques have achieved remarkable results, significantly advancing our ability to analyze and interpret complex movements from visual data. However, the common paradigm of current popular models lies in their approach to feature extraction and classification. Typically, these models employ a feature extractor followed by a 1 of N classifier, where the training process is heavily reliant on 0–1 label information [1,2,3,4,5,6]. Such a methodology, while effective in many scenarios, often falls short in capturing the rich semantic nuances of human actions. To address this gap, the concept of semantic integration into feature recognition becomes vital. This is where the CLIP (Contrastive Language–Image Pre-training) model [7] comes into play. CLIP revolutionizes the approach by emphasizing the semantic relationship between text and images, assessing the similarity between textual descriptions and visual content rather than relying solely on direct classification. The success of CLIP in diverse applications, including image classification [8,9,10,11], object detection [12], and natural language processing [13], not only demonstrates the effectiveness of semantic integration but also illuminates a promising path for its application in the domain of action recognition. By adopting a similar approach, there is a potential to enrich the feature extraction process in action recognition models, allowing for a deeper, more nuanced understanding of human actions that goes beyond mere surface-level classification.

Building upon the foundations laid by CLIP and its applications in various domains, researchers have begun exploring its potential in the realm of video understanding [14,15] and human action recognition [16,17,18,19]. The prevailing approach in these studies involves independently computing video features and category text features, and then evaluating their compatibility based on similarity measures. However, this method, while innovative, presents certain limitations specific to action recognition. The primary concern is that the computation of video features in isolation does not effectively integrate with the semantic context of the action category [16]. This results in a representation of video data that may not fully encapsulate the intricacies and semantic relevance of the actions being analyzed. Furthermore, the training tasks in these models are often singular in focus, merely assessing the match between video and text without fully exploiting the model’s capacity for nuanced recognition [14]. This approach, therefore, may not tap into the full potential of the model’s capability to discern subtle differences and similarities in complex human movements. It is these limitations that the current study seeks to address. By proposing a more integrated method of feature computation and a multifaceted training approach, this research aims to enhance the semantic alignment between action video features and action categories, thereby advancing the field of action recognition towards a more sophisticated and semantically rich understanding of human movements.

In response to these limitations of existing CLIP-inspired models in the domain of action recognition, we present ‘SkeletonCLIP++’, an enhanced model for skeleton-based human action recognition, which builds upon our earlier work with the ‘SkeletonCLIP’ model [19]. Illustrated in the model framework Figure 1, SkeletonCLIP++ is designed with three primary advancements that address the gaps in semantic action recognition:

**Weighted Frame Integration (WFI)**: As depicted in the upper portion of Figure 1, this innovation changes how action video features are computed. Moving beyond the conventional averaging of frame features, SkeletonCLIP++ introduces a weighted sum approach. Each frame’s contribution to the video feature is now determined by a weight calculated from the normalized dot product with the corresponding text features (indicated by the Weighted Frame Integration block in the diagram). This leads to a video representation that captures the semantic essence of the actions more effectively;**Contrastive Sample Identification (CSI)**: The model introduces a novel task, detailed in the lower section of Figure 1, which emphasizes the identification of the most similar negative samples—a process that improves the model’s ability to distinguish between closely resembling actions. Through the creation of [positive video, positive text], [positive video, most similar negative text], and [most similar negative video, positive text] pairs, as shown in the diagram, the model utilizes a binary classifier to discern matches, enhancing recognition precision;**BERT Text Encoder Integration**: SkeletonCLIP++ incorporates the sophisticated natural language processing capabilities of the BERT model [20] to enrich its semantic interpretation. This integration allows SkeletonCLIP++ to leverage the advanced natural language understanding of BERT, facilitating more effective learning with reduced training requirements.

These improvements aim to refine the model’s performance in capturing the nuances of human actions and provide modest but meaningful enhancements in action recognition accuracy. In theoretical terms, our enhancements to SkeletonCLIP++ focus on refining the computation of video features. By integrating these features with semantic information, the newly computed video features more accurately represent motion information relevant to action categories, which is expected to improve recognition accuracy. Furthermore, the addition of a new task to distinguish between the most similar negative samples and positive samples allows the model to achieve finer discrimination, thereby enhancing overall recognition precision. Simultaneously, the utilization of an open-source pre-trained model such as BERT helps prevent overfitting that might occur due to limited semantic data in training. Empirically, our ablation studies conducted across datasets of varying scales confirm the effectiveness of these innovations in improving the model’s accuracy. These results support the notion that integrating action recognition with semantic information is a promising direction for future research in this field. In essence, employing the CLIP philosophy in action recognition is not only valid but also opens avenues for further exploration and development in combining deep semantic understanding with dynamic action interpretation.

## 2. Related Works

### 2.1. Methods for Human Action Recognition

In the realm of deep learning, the pursuit of human action recognition began with the deployment of 2D CNNs [21,22], adept at processing spatial features within images [2,3]. Karpathy et al. [2] primarily innovated in providing an extensive empirical evaluation of CNNs on large-scale video classification, utilizing a new dataset of 1 million YouTube videos to demonstrate the effectiveness of CNNs in accurately classifying a vast array of video content. As research progressed, 3D CNNs were introduced to capture the temporal dynamics within video data, enhancing the comprehension of action sequences over time [1,23,24]. Tran et al. [23] proposed an effective approach for spatio-temporal feature learning using deep 3-dimensional convolutional networks (C3D) for enhanced analysis. Carreira and Zisserman [1] introduced a significant advancement by re-evaluating the state-of-the-art architectures. They inflated 2D kernel and initialized the model with the pre-trained 2D CNN net to reach better recognition performances. However, the reliance on large annotated datasets posed a challenge for these methods.

Graph Convolutional Networks [5,25,26,27] (GCNs) later emerged to more adeptly handle non-Euclidean data, such as skeletal information, capturing the intricate interplay of human motions and joint relationships. Yan et al. [5] introduced a noval model called Spatial-Temporal Graph Convolutional Networks (ST-GCN), which innovatively addressed skeleton-based human action recognition by effectively capturing both spatial and temporal patterns in dynamic skeletons using graph convolutional networks. Follow-up works by Shi and Zhang [26] enhanced the model ST-GCN by introducing a novel two-stream adaptive graph convolutional network (2s-AGCN) that adapts the graph topology dynamically for more effective skeleton-based action recognition. Liu et al. [27] improves upon the ST-GCN model by focusing on disentangling and unifying graph convolutions, which allows for a more efficient and comprehensive understanding of action recognition task. To address the long-term dependencies present in video and sequence data, LSTMs [28,29,30] were widely adopted to construct temporal relationships with different spatial alignments. More recently, the Transformer model [20], celebrated for its success in natural language processing, made its foray into the visual domain. Its self-attention mechanism [31] is particularly effective at capturing global dependencies, a vital attribute for complex action recognition tasks. Li et al. proposed an innovative group activity recognition network [6] that effectively captures spatial-temporal contextual information through a clustered Transformer approach, enhancing the understanding of group dynamics. Spatial-Temporal Transformer network (ST-TR) [32] models dependencies between joints using the Transformer architecture, offering a significant improvement in capturing complex human movements.

This evolutionary trajectory reflects a continuous ambition within the deep learning community to craft more precise and interpretative action recognition systems. Each advancement has supplemented and optimized the capabilities of prior techniques, striving to enhance the representational power and recognition accuracy of models. From the localized feature extraction of 2D CNNs to the spatio-temporal comprehension afforded by 3D CNNs, the structural data processing by GCNs, the long-term sequence handling by LSTMs, and up to the global contextual relationships captured by Transformers, each step has represented a further enhancement of action recognition capabilities. These developments not only signify technological progress but also indicate potential future research directions, such as multimodal learning and adversarial training, aligning with the central tenets of our work. In our study, we utilize a video encoder based on 3D CNNs, in line with the PoseConv3D model [4] for fair comparison. Our model, named SkeletonCLIP++, innovates beyond their approach by implementing a weighted frame integration technique, which departs from global average pooling to highlight the significance of key frames in action sequences, leveraging semantic information through an adversarial learning approach, in keeping with the current research trajectory.

### 2.2. Advancements in Natural Language Processing

The field of natural language understanding has witnessed significant advancements with the advent of the Transformer model, particularly due to its self-attention mechanism. Unlike traditional recursive [33,34] or convolutional structures [35], self-attention enables the model to simultaneously consider all other words in a sentence when processing each word, capturing intricate inter-word relationships with ease. This mechanism directly addresses long-range dependencies, overcoming the limitations previously encountered by sequential models. The innovation of BERT (Bidirectional Encoder Representations from Transformers) [20] lies in its bidirectional context learning. By employing two pre-training tasks, Masked Language Modeling (MLM) and Next Sentence Prediction (NSP), BERT learns comprehensive word representations. These tasks are designed to imbue the model with an understanding of language structures, preparing it to tackle complex linguistic constructs more effectively.

BERT’s effectiveness has been proven across various NLP tasks, showcasing its versatility and superior performance in semantic understanding [36], sentiment analysis [20], and question-answering systems [37]. Zhang et al. [36] enhances BERT’s language understanding capabilities by integrating semantic information, evaluated across multiple NLP tasks including natural language inference and text classification, thereby improving the model’s ability to grasp deeper semantic relationships. Koroteev [37] highlights BERT’s application in question-answering systems, showcasing its ability to enhance the accuracy and relevance of responses by deeply understanding the context and nuances of natural language queries. Its ability to extract profound semantic features from text has prompted researchers to explore its application in the visual domain, particularly for semantic understanding of videos and images. The success in these tasks highlights BERT’s extensive application potential and its achievement in yielding remarkable results.

Owing to the extensive pre-training on large datasets, BERT has become an invaluable tool to reduce the training load and prevent overfitting, especially when textual information is sparse in the application process. Therefore, in this study, we opt for a pre-trained BERT model as our text encoder, leveraging its pre-training to enhance our network’s performance. This section has laid the groundwork for the subsequent section, which will delve into the specifics of applying BERT within the visual domain, marking a cross-disciplinary effort to unify the understanding of both textual and visual modalities under a cohesive learning framework.

### 2.3. Applications of the CLIP Model

Following the significant achievements of BERT in natural language processing, researchers have extended its methodologies to the visual domain, particularly in the semantic understanding of images and videos. Among the most notable outcomes of interdisciplinary endeavors is the CLIP model. CLIP’s innovation lies in its ability to learn visual concepts from natural language supervision, effectively bridging the gap between vision and language understanding. The model operates on the principle of contrasting text–image pairs, learning to associate images with captions in a manner that generalizes to a wide array of visual tasks. This has led to exceptional performance in various image-related applications, such as zero-shot classification, object detection, and even complex scene understanding.

In the realm of video understanding, CLIP’s principles have been adapted to leverage the temporal inherent in videos. Researchers approach videos as sequences of images, extending the use of Vision Transformers (ViT) [38] as video encoders. Some methodologies [14,16,17,18] employ the output of the ‘**cls_token**’ as the feature representation for image frames, while others [15,19] average all patch output features. Transitioning from frame features to video features, several models [14,15,18,19] average frame features directly; others [16,17] fuse temporal information using LSTM or Transformer structures before averaging to capture video features. Innovations such as the introduction of a message token for temporal feature extraction and fusion layers within ViT models have also been explored to enhance temporal understanding [17]. Focusing on prompting schemes has also been a strategy to fine-tune models for specific video understanding tasks, leveraging the pre-trained nature of these models to improve performance with minimal additional training. Vita-CLIP [18] introduced a novel multi-modal prompt learning scheme that effectively balanced supervised and zero-shot performance in action recognition, leveraging the strengths of CLIP in a unified framework for both video and text modalities.

However, a common thread among these studies is the use of ViT as the video encoder, owing to the rich image data available for pre-training. Yet, for the domain-specific task of action recognition using skeleton data, ViT does not capitalize on its pre-training advantage due to the distinct nature of the data. In our work, we select a 3D CNN as the video encoder, similar to our previous work SkeletonCLIP [19], which has shown robust performance on skeleton-based action recognition tasks. Our model, unlike most current methodologies, does not employ global average pooling for frame feature to video feature computation. Instead, we opt for a weighted integration approach, where each frame’s weight is correlated with the textual features, allowing for a semantic-rich video feature conducive to accurate recognition tasks. Additionally, our work introduces the task of contrastive sample identification, challenging a binary classifier to discern closely resembling samples. This novel task aims to enhance the model’s discriminative capacity, ultimately improving overall recognition performances.

In summarizing the related work within this chapter, we have traced the evolution of techniques from vision-based models to language models and their eventual convergence in our current research. Our work with the SkeletonCLIP++ model seeks to assimilate these developments, offering improvements on existing methods in skeleton-based human action recognition.

Our approach utilizes the proven efficacy of 3D CNNs for video encoding, informed by prior research, and enhances feature computation by integrating semantic context—a method not yet widely adopted in current frameworks. We introduce a discriminative task to refine the model’s ability to distinguish between closely resembling actions. While we build upon the established foundation set by BERT and CLIP, our model is an extension rather than a reinvention, aiming to optimize the synergy between textual and visual modalities in action recognition. The incorporation of adversarial learning tasks reflects a step towards improving the nuanced understanding of complex actions.

In the next section, we will introduce the specifics of our model, particularly focusing on the implementation details of the Weighted Frame Integration (WFI) module and the Contrastive Sample Identification (CSI) task. Additionally, we will present a thorough description of the training process.

## 3. Methods

In this section, we will outline the architecture and implementation details of our SkeletonCLIP++ model, focusing particularly on its two main components: Weighted Frame Integration (WFI) module Section 3.1 and Contrastive Sample Identification (CSI) task Section 3.2. This section is designed to provide a detailed explanation of how these features are conceptualized in our model framework.

Firstly, the WFI module represents a significant advancement in our model’s ability to process and interpret video data. We will delve into how this module applies a weighted mechanism to frame features, factoring in the semantic correlation with corresponding text features. Subsequently, we will turn our attention to the CSI task, a novel approach to training our model. We will detail the methodology behind identifying the most similar negative samples. Through these sections, we aim to provide a comprehensive understanding of the underlying mechanics of our model, highlighting how each component contributes to the overall efficacy of action recognition in the SkeletonCLIP++ framework.

### 3.1. Weighted Frame Integration

In this section, we explain the implementation of the Weighted Frame Integration module, a main contribution of our SkeletonCLIP++ model that innovatively computes video features by combining frame features with corresponding textual semantics.

For an action sequence spanning *T* frames, we first encode the input to extract a vector of frame features, $e_{f}=\left[f_{1},\ldots ,f_{T}\right]$, through the video encoder. Simultaneously, we obtain a text feature vector, $e_{x}$, by passing an action category description, enhanced with the ‘**cls_token**’, through the BERT text encoder. Then, we use our WFI block to calculate the video feature $e_{v}$, as illustrated in Figure 2. The essence of the WFI module is to calculate a similarity score, $s_{i}$, for each frame feature $f_{i}$ by taking the dot product with the text feature vector $e_{x}$. These scores are then normalized using a softmax function to yield a set of weights $a=[a_{1},\ldots ,a_{T}]$, reflecting the semantic relevance of each frame to the textual description. The final video feature, $e_{v}$, emerges as a weighted sum of the frame features, where the weights are directly influenced by the semantic alignment of each frame with the text feature. This process ensures that frames which are semantically more significant to the described action are accentuated in the resulting video feature representation. The following pseudo code provides a succinct depiction of the WFI module’s operations:

In summary, the WFI module embodies our endeavor to advance the semantic processing capabilities within the action recognition domain, integrating contextual understanding directly into the feature extraction phase. This approach not only aligns with, but also extends the current state-of-the-art methodologies by infusing a deeper level of semantic reasoning into the video encoding process.

### 3.2. Contrastive Sample Identification

In this section, we address the implementation details of the Contrastive Sample Identification (CSI) task within our SkeletonCLIP++ model, as illustrated in the accompanying schematic Figure 3. The CSI task is central to enhancing the model’s discriminative power by training it to ascertain whether a fused feature originates from a matching video–text pair. The following pseudo code provides a forward process of the CSI task. We will give a detailed description of this task.

For each video—hereafter referred to as the positive video sample—we identify the corresponding positive text sample anticipated to match and also seek out the most similar negative text sample. Similarly, for each category text, we obtain the matching positive video sample and the most similar negative video sample. The selection of the most similar negative sample is quantified by identifying the sample with the maximum similarity score compared to the positive sample, formulated in Equation (Equation 1):(1)$$most\_sim\_neg=\underset{x\in neg\_samples}{\arg\max}Similarity\left(pos\_samples,x\right)$$

Consequently, we obtain three types of sample pairs: [positive video sample, positive text sample], [positive video sample, most similar negative text sample], and [most similar negative video sample, positive text sample]. Each pair is processed to compute its feature representation. The fusion of features from the two modalities is achieved using the same BERT model that processes text features. The text features, enhanced with ‘**enc_token**’, are taken as the input of text encoder. The video features are input into the Cross Attention module of encoder, and the output of **‘enc_token’** serves as the fused feature, which is then fed into a binary classifier. The classifier is trained under the premise that only the first pair constitutes a match, while the latter two are mismatches. This training strategy is designed to refine the model’s ability to distinguish between closely related categories.

The fusion method employs the BERT model’s output as the fused feature for classification. By incorporating the textual feature as input and the video feature in the Cross Attention module, the model leverages BERT’s sophisticated encoding capabilities to generate a feature representation conducive to binary classification. This task is integral to our model, aiming to enhance the nuanced differentiation between similar action categories, thereby improving overall recognition performance.

### 3.3. Forward Process and Loss Function Calculation

In this section, we elucidate the forward process of our model for a batch of data and the methodology employed in computing the Loss function.

**Forward Process:** The forward process begins with the input of a batch of video $\left[V_{1},\ldots ,V_{B}\right]$ and corresponding textual descriptions $\left[L_{1},\ldots ,L_{B}\right]$ into the model. Each video is processed through a video encoder, and each textual description through a text encoder, resulting in a set of frame features $\left[e_{f1},\ldots ,e_{fB}\right]$ and text features $\left[e_{x1},\ldots ,e_{xB}\right]$, respectively. These features are the foundational elements for both the Semantic Similarity Matching (SSM) and Contrastive Sample Identification (CSI) tasks.

The frame features and text features are first passed through the WFI module. This process generates refined video features $\left[e_{v1},\ldots ,e_{vB}\right]$ that encapsulate the semantic essence of the actions more effectively. The detailed process has been shown in Algorithm 1. In parallel, for the CSI task, the model utilizes the output of the WFI module to select the most similar negative text and video features, creating three types of feature pairs: [positive video feature, positive text feature], [positive video feature, most similar negative text feature], and [most similar negative video feature, positive text feature]. These pairs are then integrated within the CSI module and fed into a binary classifier to assess match probability, thereby enhancing the model’s ability to distinguish between closely related action categories. Algorithm 2 shows the detailed process.
**Algorithm 1** Weighted Frame Integration (WFI) Module**Require:** Frame features $e_{f}=\left[f_{1},\ldots ,f_{T}\right]$, Text feature $e_{x}$**Ensure:** Video feature $e_{v}$  1: s←[]                                                                        ▹ Initialize similarity scores array  2: **for** 
i←1 
**to** 
*T* 
**do**  3:     $s_{i}\leftarrow \mathrm{dot}\_\mathrm{product}\left(f_{i},e_{x}\right)$  4:     Append $s_{i}$ to *s*  5: **end for**  6: a←softmax(s)                                                ▹ Normalize scores to obtain weights  7: $e_{v}\leftarrow \mathrm{dot}\_\mathrm{product}\left(a,e_{f}\right)$                        ▹ Compute weighted sum of frame features  8: **return** 
$e_{v}$
**Algorithm 2** Contrastive Sample Identification (CSI) Task**Require:** Positive video features $V^{+}$, Positive text features $T^{+}$, Negative text features $T^{-}$, Negative video features $V^{-}$**Ensure:** Binary classification of fused features as matching or mismatching  1: **function** FindMostSimilarNegativeSample($V^{+}$, $T^{-}$, $T^{+}$)  2:     $similarity_{max}\leftarrow -\infty$  3:     $neg\_sample_{max}\leftarrow \mathrm{null}$  4:     **for** each $t^{-}$ in $T^{-}$ **do**  5:         $similarity\leftarrow \mathrm{Similarity}(V^{+},t^{-})$  6:         **if** $similarity>similarity_{max}$ **then**  7:            $similarity_{max}\leftarrow similarity$  8:            $neg\_sample_{max}\leftarrow t^{-}$  9:         **end if**10:     **end for**11:     **return** $neg\_sample_{max}$12: **end function**13: **function** FuseFeatures(*v*, *t*)14:     $t_{enc}\leftarrow$ AddEncToken(*t*)                                    ▹ Prepend ‘**enc_token**’ to text feature15:     $f\leftarrow \mathrm{BERT}(t_{enc},v)$            ▹ Input text to BERT and video feature to Cross Attention16:     **return** *f*                                                ▹ Output from BERT corresponding to enc_token17: **end function**18: **function** ContrastiveSampleIdentification($V^{+}$, $T^{+}$, $T^{-}$, $V^{-}$)19:     $P_{match}\leftarrow \left[\right]$20:     $P_{mismatch}\leftarrow \left[\right]$21:     $neg\_text\leftarrow \mathrm{FindMostSimilarNegativeSample}(V^{+},T^{-})$22:     $neg\_video\leftarrow \mathrm{FindMostSimilarNegativeSample}(T^{+},V^{-})$23:     $fused_{match}\leftarrow \mathrm{FuseFeatures}\left(V^{+},T^{+}\right)$24:     $fused_{mismatch\_text}\leftarrow \mathrm{FuseFeatures}\left(V^{+},neg\_text\right)$25:     $fused_{mismatch\_video}\leftarrow \mathrm{FuseFeatures}\left(neg\_video,T^{+}\right)$26:     Append $fused_{match}$ to $P_{match}$27:     Append $fused_{mismatch\_text}$ and $fused_{mismatch\_video}$ to $P_{mismatch}$28:     **for** each fused_feature in $P_{match}\cup P_{mismatch}$ **do**29:         classification←Classifier(fused_feature)30:         **if** fused_feature is in $P_{match}$ **then**31:            Expect classification to be 132:         **else**33:            Expect classification to be 034:         **end if**35:     **end for**36: **end function**

**Loss Function:** The model employs a dual-component Loss function, integrating the losses from both the SSM and CSI tasks to guide the training process:KL Loss for SSM Task: The Kullback–Leibler (KL) divergence loss is used for the SSM task, quantifying the similarity between the video features and the text features. This loss encourages the model to accurately match video and text pairs based on their semantic content;Cross Entropy Loss for CSI Task: For the CSI task, the Cross Entropy Loss is calculated based on the binary classification of the feature pairs as matching or non-matching. This loss component is crucial for enhancing the model’s ability to discriminate between closely related actions by learning from the most challenging negative samples.

The overall network loss is the sum of the KL Loss from the SSM task and the Cross Entropy Loss from the CSI task. This combined Loss function is pivotal in simultaneously optimizing the model for both semantic similarity matching and contrastive sample identification, ensuring a comprehensive understanding and differentiation of human actions.

## 4. Experiments

In this section, we will validate our model through various experiments. We will present a series of tests designed to evaluate the efficacy of SkeletonCLIP++ across various datasets. Our experiments are crafted to offer a transparent and comprehensive appraisal of the model’s performance. Through these experiments, we anticipate demonstrating the practical value and the significant strides our model represents in the field of action recognition.

### 4.1. Datasets

In the realm of human action recognition, we evaluate on three datasets: HMDB-51 [39], UCF-101 [40] and NTU RGB+D 60 [41]. These datasets are selected for their widespread use and the comprehensive nature of their action categories, which provide a robust foundation for validating the efficacy of our SkeletonCLIP++ model.

**HMDB-51**: This dataset contains 6766 video clips distributed across 51 action categories. It is known for its diversity, incorporating actions from various sources and capturing a wide range of human activities. The dataset is traditionally split into three distinct sets: split1, split2, and split3. Each split comprises a unique arrangement of training and test sets, ensuring that models are assessed on varied samples.

**UCF-101**: Comprising 13,320 videos spanning 101 action categories, UCF-101 is another widely-utilized dataset for action recognition tasks. Similar to HMDB-51, it features three splits, providing different training and testing scenarios to evaluate the generalizability and robustness of recognition algorithms.

**NTU RGB+D 60**: Containing 60 action categories with 56,880 video samples, NTU RGB+D 60 is one of the largest datasets for human action recognition. It is especially notable for its incorporation of RGB videos, depth map sequences, and 3D skeleton data. The dataset can be divided into cross-subject (X-Sub) and cross-view (X-View) evaluation criteria, offering distinct challenges based on subject variations and camera angles, respectively.

In our experiments, each dataset’s split configuration is utilized to thoroughly assess the recognition precision of our model. Notably, our model demonstrates a more pronounced improvement in recognition accuracy on the smaller-scale datasets, HMDB-51 and UCF-101. This suggests that the innovations introduced in our model, particularly the Weighted Frame Integration and Contrastive Sample Identification tasks, are highly effective even when data is limited in volume.

The choice to use datasets formatted similarly to those in the PoseConv3D study [4] allows for a fair comparative analysis, highlighting the advancements contributed by our model. Given that these datasets are composed of 2D skeleton structure extractions from video footage, they are well-suited for tasks focusing on skeleton-based human action recognition. This alignment with the PoseConv3D study ensures that any observed improvements can be attributed to our model’s innovative components rather than differences in data preprocessing or format. The subsequent experimental results will offer insights into the model’s performance across these diverse and complex datasets, proving the validity of our proposed model.

### 4.2. Experimental Settings

Our proposed SkeletonCLIP++ model is implemented using the Pytorch framework, capitalizing on its dynamic graph construction and intuitive design for deep learning research. The training is conducted on 2 NVIDIA 3090Ti GPUs, which provide the computational power necessary to handle the intensive demands of training sophisticated models on large datasets.

To ensure consistency and fairness in our experiments, all models are trained with identical initial learning rates and follow the same learning rate adjustment strategy. This uniform approach guarantees that any observed differences in performance are due to the model architectures themselves rather than discrepancies in training procedures. For the text encoding component of our model, we employ the ’bert-base-uncased’ variant of the BERT pre-trained model. This version of BERT, which does not distinguish between uppercase and lowercase text, is widely used for its balance between size and performance, making it an ideal choice for integrating robust natural language understanding into our framework. In our comparative analysis, we benchmark the performance of SkeletonCLIP++ against a suite of state-of-the-art models. These models are evaluated using their publicly available source codes, ensuring that the comparison is conducted under equivalent conditions.

This section sets the stage for the presentation of our empirical findings. In the following sections, we will detail the performance metrics, discuss the outcomes of our experiments, and provide insights into the strengths and limitations of SkeletonCLIP++ as illuminated by our rigorous experimental evaluation. Our implementation details are available at https://github.com/eunseo-v/skeletonclipPLUS (accessed on 31 January 2024).

### 4.3. Ablation Study

In this section, we conduct a series of ablation studies to analyze and validate the individual contributions of the distinct components integrated into our SkeletonCLIP++ model. Each experiment is carefully crafted to isolate the effects of specific modules and configurations, providing empirical evidence that supports our theoretical assertions regarding the model’s performance enhancements.

#### 4.3.1. Effect of the WFI Module

To ascertain the efficacy of the WFI module within our SkeletonCLIP++ framework, we conducted an ablation study focused on the similarity computation between video and text features. Essential to this process is the mapping of both feature types into a common dimensional space. We evaluated two distinct mapping approaches: a linear mapping and a residual non-linear mapping. The linear mapping was implemented using a straightforward linear function. In contrast, the residual non-linear mapping comprised a direct mapping component and a residual connection. The direct mapping utilized two linear layers interspaced with a GELU non-linearity to achieve a non-linear transformation, while the residual connection employed a linear function to facilitate gradient flow. For the conversion of frame similarity scores into weights, we experimented with two activation functions: the sigmoid and the softmax function. Our objective was to identify the most effective combination through empirical evaluation.

The experiments were carried out on the NTU RGB+D 60 dataset. To optimize training time, the dataset was not repeated during model training. The experimental results are summarized in Table 1.

The results indicate that employing a linear mapping function in conjunction with the softmax activation function yields the highest recognition accuracy. Consequently, this configuration was adopted for all subsequent experiments within our research. An analysis of the results suggests that the linear mapping function’s simplicity is perhaps better suited to our model’s architecture, avoiding the potential overfitting that might occur with more complex mappings. The softmax function’s effectiveness could be attributed to its probabilistic interpretation, which seems to be more aligned with the task of weighing frame importance based on semantic similarity. In subsequent experiments, the WFI module consistently utilizes a linear mapping function coupled with the softmax activation function for implementation.

To evaluate the impact of the Weighted Frame Integration (WFI) module on recognition accuracy, we conducted comparative experiments on the HMDB-51 and UCF-101 datasets, using their respective split1, split2, and split3 criteria. The results of these experiments are presented in Table 2, showcasing the effectiveness of the WFI module in enhancing model performance.

The table clearly indicates an increase in recognition accuracy across all dataset splits with the incorporation of the WFI module. The most significant improvement, exceeding 3%, is observed in HMDB-51 split1 and UCF-101 split3. This improvement highlights the efficacy of employing a weighted approach in constructing video features from frame-level data. By attributing variable importance to different frames based on their semantic relevance, the WFI module evidently enhances the model’s ability to generate more accurate representations of the actions depicted in the videos. These results underscore the value of integrating semantically-informed weighting mechanisms into action recognition models, particularly in scenarios where nuanced understanding of action sequences is critical. Such enhancements are pivotal in advancing the state-of-the-art in human action recognition, as they offer a more refined approach to capturing the essence of complex activities.

#### 4.3.2. Effect of the CSI Task

To examine the efficacy of incorporating the Contrastive Sample Identification (CSI) task in our SkeletonCLIP++ model, we conducted ablation experiments on both HMDB-51 and NTU RGB+D 60 datasets. These experiments were designed to ascertain the impact of the CSI task on the overall recognition accuracy, especially in distinguishing similar action categories. We structured our experiments across different splits of the HMDB-51 dataset and the two common evaluation methods of the NTU RGB+D 60 dataset. The results have been shown in Table 3.

These results indicate a modest but consistent improvement in recognition accuracy upon adding the CSI task across all dataset splits. It is important to note that while the CSI task is designed to enhance the model’s ability to differentiate between closely resembling actions, it does not necessarily lead to a significant increase in overall dataset accuracy. This is due to the fact that for many action categories, the most similar negative samples are still quite distinct from the positive samples, making them relatively easy for the classifier to differentiate. The CSI task’s contribution, therefore, lies more in its ability to refine the model’s discernment capabilities in nuanced scenarios, rather than in broadly increasing accuracy across all categories. These findings suggest that the task is particularly beneficial in complex recognition scenarios where subtle differences between actions are critical.

#### 4.3.3. Effect of Pre-Trained BERT Model

To validate the effectiveness of utilizing BERT as a text encoder in our SkeletonCLIP++ model, we conducted an ablation study focusing on different configurations of the BERT model. The experiment aimed to ascertain the impact of pre-training and fine-tuning on recognition accuracy. We compared three scenarios: using a BERT model without pre-training, a pre-trained BERT model with frozen weights, and a pre-trained BERT model with trainable (fine-tuned) weights. The experiments were performed on the NTU RGB+D 60 dataset, and the results are as follows:**No Pre-training**: Recognition accuracy was 89.42%;**Pre-trained with Frozen Weights**: Recognition accuracy improved to 91.91%;**Pre-trained with Fine-tuning**: Achieved the highest recognition accuracy at 93.05%.

These results align with our expectations. Given BERT’s extensive parameterization, training it from scratch can easily lead to overfitting. In contrast, utilizing a pre-trained model and fine-tuning it allows for more accurate text feature extraction, which better complements the corresponding video features. It was also noted during experimentation that a lower initial learning rate is beneficial when fine-tuning the BERT model, contributing to better recognition performance.

The following bar chart in Figure 4 visually represents these findings.

### 4.4. Comparison with Other Methods

To demonstrate the superiority of our proposed SkeletonCLIP++ model, we conducted a comparative analysis against three related methods: ActionCLIP [16], PoseConv3D [4], and SkeletonCLIP [19]. This comparison was performed across various splits of the HMDB-51, UCF-101, and NTU RGB+D 60 datasets. The recognition accuracy of each method under different dataset splits is presented in Table 4.

The results clearly indicate that SkeletonCLIP++ outperforms the other methods in all test cases, particularly on the smaller-scale datasets HMDB-51 and UCF-101, where significant improvements in recognition performance were observed. Our analysis suggests that the WFI module in SkeletonCLIP++ enhances the consistency between video features and text features of matching sample pairs. The CSI task contributes to the model’s improved ability to recognize similar action categories in limited data scenarios. Moreover, employing a pre-trained BERT model as the text encoder mitigates overfitting, leading to more accurate and semantically rich text feature extraction. These findings collectively validate the effectiveness of the integrated components in SkeletonCLIP++, showcasing its superiority in the field of action recognition.

We also analyze the computational efficiency of our proposed SkeletonCLIP++ model in comparison with two baseline models, ActionCLIP and SkeletonCLIP, across three benchmark datasets: HMDB-51, UCF-101, and NTU RGB+D 60. Our experimental setup recorded the training time for one epoch across the mentioned datasets and models. The results are shown in Table 5.

These findings indicate that SkeletonCLIP++ requires a modest increase in training time compared to SkeletonCLIP and is comparable to ActionCLIP, especially on smaller datasets such as HMDB-51 and UCF-101. The slightly longer training time for SkeletonCLIP++ on the NTU RGB+D 60 dataset is justified by the significant enhancements in model performance, as detailed below.

To assess the impact of the additional training time on model performance, we analyzed the loss function decay and the validation set accuracy improvement over training epochs for the HMDB-51 and UCF-101 datasets. Results have been shown in Figure 5 and Figure 6.

The analysis revealed that SkeletonCLIP++ not only converges faster but also achieves higher recognition accuracy compared to the baseline models. This is a testament to the effectiveness of the integrated Weighted Frame Integration (WFI) module and the Contrastive Sample Identification (CSI) task in enhancing the model’s semantic understanding and discriminative capacity. The graphical analysis clearly shows that SkeletonCLIP++ exhibits a steeper decline in the loss function and a more rapid improvement in validation accuracy across epochs. This accelerated convergence, coupled with superior final accuracy, underscores the value of the proposed modifications in the SkeletonCLIP++ model.

The modest increase in training time for SkeletonCLIP++ is a worthwhile trade-off for the notable improvements in action recognition accuracy. This is particularly true for applications where model performance is critical, and a slight extension in training duration is acceptable. The efficiency of SkeletonCLIP++ in smaller datasets such as HMDB-51 and UCF-101, where the absolute increase in training time is minimal, further supports its practicality. Moreover, the model’s ability to achieve faster convergence means that the overall training duration may not significantly exceed that of the baselines, especially when considering the reduced number of epochs needed to reach optimal performance.

In conclusion, our SkeletonCLIP++ model, with its innovative approach to video feature computation and the addition of a novel training task, demonstrates a compelling balance between computational time and action recognition accuracy. The empirical evidence supports the feasibility of our approach, highlighting its potential for advancing the state-of-the-art in human action recognition.

## 5. Conclusions

In this paper, we presented SkeletonCLIP++, a refined approach for human action recognition that integrates advanced video and text feature analysis techniques. The model’s key innovations—Weighted Frame Integration (WFI), Contrastive Sample Identification (CSI), and the integration of the BERT text encoder—collectively enhance its ability to interpret and classify complex actions accurately.

The WFI module marks a significant departure from traditional methods of averaging frame features, adopting a weighted sum approach that assigns variable importance to frames based on their semantic relevance. This advancement leads to a more nuanced and semantically rich video representation. The CSI task introduces a novel strategy for distinguishing between closely resembling actions by focusing on the identification and utilization of the most similar negative samples. This approach significantly improves the model’s precision in differentiating subtle action variations. Furthermore, the integration of the pre-trained BERT model as the text encoder enriches the model’s semantic interpretation capabilities, leveraging advanced natural language processing to facilitate more effective and efficient learning.

Empirical evaluations across multiple datasets, including HMDB-51, UCF-101, and NTU RGB+D 60, demonstrate that SkeletonCLIP++ achieves superior performance, especially on smaller-scale datasets. This improvement underscores the efficacy of our model’s innovative components in enhancing action recognition capabilities. The combination of these advancements in SkeletonCLIP++ represents a thoughtful contribution to the field of human action recognition, offering a model that not only achieves high accuracy but also provides a deeper understanding of the intricate relationship between video and textual data in representing human actions.

## Figures and Tables

**Figure 1 sensors-24-01189-f001:**
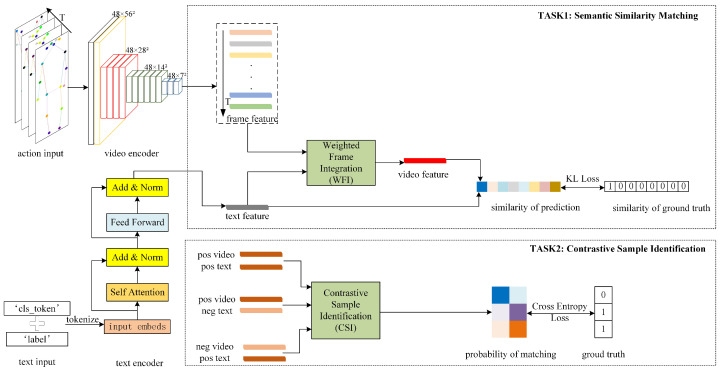
Overview of the SkeletonCLIP++ architecture for skeleton-based human action recognition. The process begins with an action input passing through a video encoder to compute frame features. These features are then integrated using a Weighted Frame Integration (WFI) block, where weights are derived from the dot product with text features from the BERT text encoder. The model also introduces a Contrastive Sample Identification (CSI) task that discriminates between positive samples and the most similar negative samples. The combined features are fed into a binary classifier to determine the probability of matching, facilitating enhanced recognition performance.

**Figure 2 sensors-24-01189-f002:**
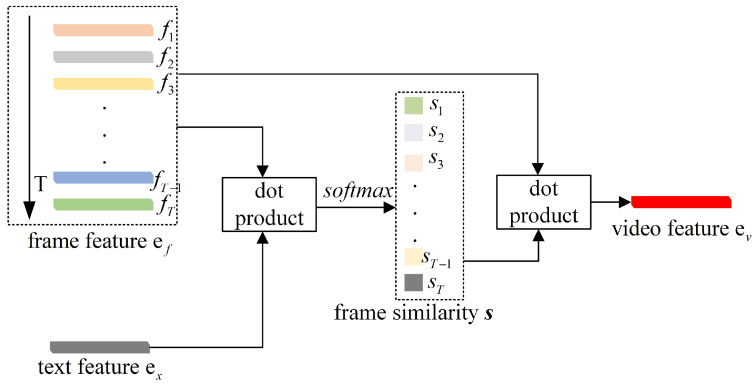
Schematic representation of the Weighted Frame Integration (WFI) module in the SkeletonCLIP++ framework. This diagram illustrates the transition from individual frame features to a unified video feature vector that captures the semantic essence of the depicted action, facilitating enhanced recognition accuracy in the subsequent stages of the SkeletonCLIP++ model.

**Figure 3 sensors-24-01189-f003:**
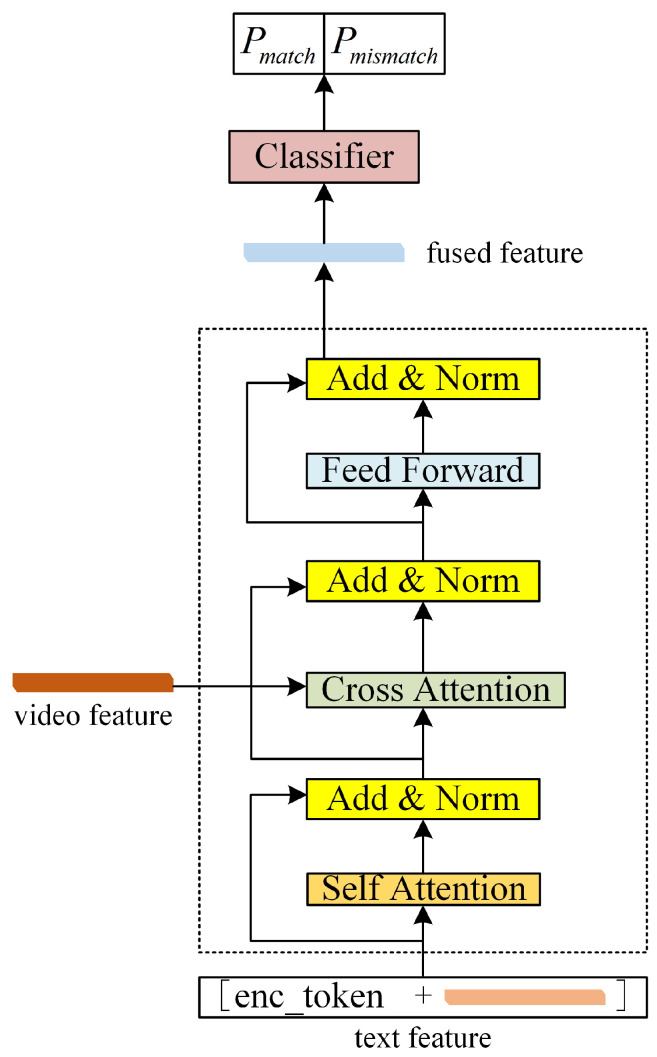
Schematic representation of the Contrastive Sample Identification (CSI) task in the SkeletonCLIP++ framework.

**Figure 4 sensors-24-01189-f004:**
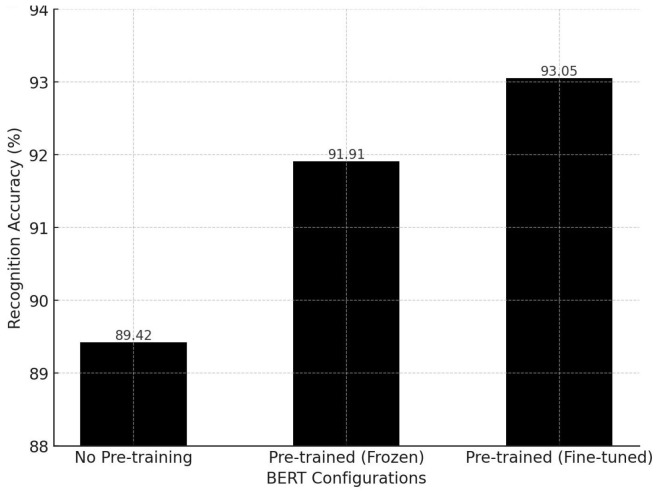
Recognition accuracy across different BERT configurations on the NTU RGB+D 60 dataset.

**Figure 5 sensors-24-01189-f005:**
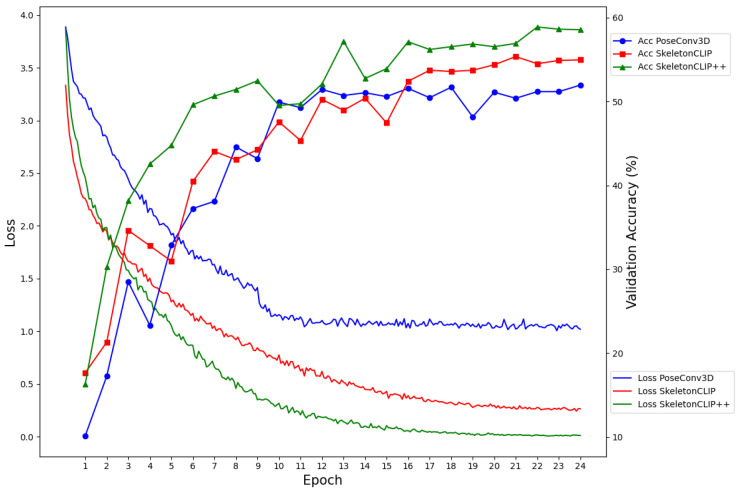
Convergence and accuracy results using different models on HMDB-51 dataset.

**Figure 6 sensors-24-01189-f006:**
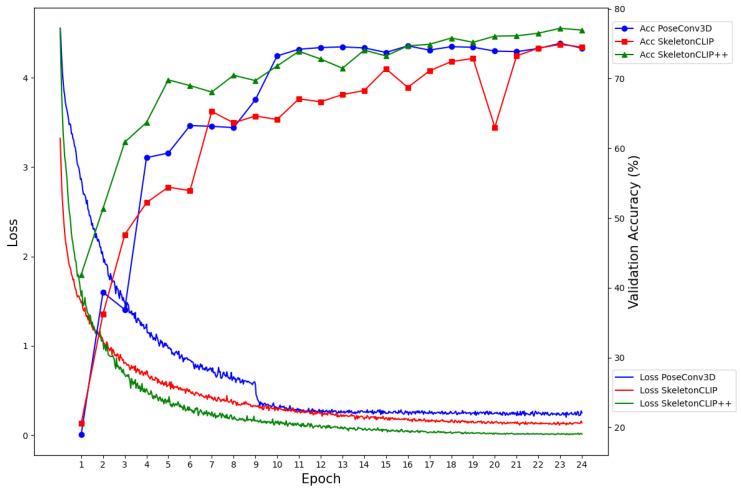
Convergence and accuracy results using different models on the UCF-101 dataset.

**Table 1 sensors-24-01189-t001:** Recognition accuracy with different mapping functions and activation functions on the NTU RGB+D 60 dataset.

Mapping Function	Activation Function	Accuracy (%)
Non-linear	Sigmoid	88.82
Non-linear	Softmax	89.62
Linear	Sigmoid	89.04
Linear	Softmax	**90.06**

**Table 2 sensors-24-01189-t002:** Recognition accuracy improvement with the WFI module on HMDB-51 and UCF-101 dataset splits.

Dataset/Split	Without WFI (%)	With WFI (%)	Improvement (%)
HMDB-51/split1	55.36	58.89	+3.53
HMDB-51/split2	54.97	56.99	+2.02
HMDB-51/split3	52.68	54.51	+1.83
UCF-101/split1	74.89	77.21	+2.32
UCF-101/split2	77.13	79.22	+2.09
UCF-101/split3	76.70	79.76	+3.06

**Table 3 sensors-24-01189-t003:** Impact of CSI task on recognition accuracy in HMDB-51 and NTU RGB+D 60 datasets.

Dataset/Split	Without CSI (%)	With CSI (%)	Improvement (%)
HMDB-51/split1	55.36	56.27	+0.91
HMDB-51/split2	54.97	55.16	+0.19
HMDB-51/split3	52.68	55.62	+2.94
NTU RGB+D/X-Sub	92.32	93.05	+0.73
NTU RGB+D/X-View	95.44	96.28	+0.84

**Table 4 sensors-24-01189-t004:** Recognition performance comparison with several related methods on three mainstream datasets.

Dataset/Split	ActionCLIP (%)	PoseConv3D (%)	SkeletonCLIP (%)	SkeletonCLIP++ (%)
HMDB51/split1	49.59	51.96	55.36	**59.82**
HMDB51/split2	49.86	52.61	54.97	**57.18**
HMDB51/split3	48.04	49.02	52.68	**57.47**
UCF-101/split1	62.89	75.05	74.89	**78.89**
UCF-101/split2	64.65	75.76	77.13	**80.41**
UCF-101/split3	65.84	76.70	76.70	**80.92**
NTU60/X-Sub	85.96	93.09	92.32	**93.39**
NTU60/X-View	84.33	96.03	95.44	**96.47**

**Table 5 sensors-24-01189-t005:** Time consumption per epoch using different action recognition models across three datasets.

Dataset/Split	ActionCLIP (min)	SkeletonCLIP (min)	SkeletonCLIP++ (min)
HMDB51/split1	12	10	11
UCF-101/split1	33	23	28
NTU60/X-Sub	100	100	120

## Data Availability

The dataset presented in this study is available. The dataset download link is located in the dataset part of website https://github.com/eunseo-v/skeletonclipPLUS.git.

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
