# Peer review of "Advancing Human Motion Recognition with SkeletonCLIP++: Weighted Video Feature Integration and Enhanced Contrastive Sample Discrimination"

_sensors, 2024, doi:10.3390/s24041189_

Round 1

Reviewer 1 Report

Comments and Suggestions for Authors

The paper is well written and organized. The authors proposed a new model and increased the accuracy of the existing models.
However, as a reviewer I think there are some additions need to be done. The authors must add the images of the human action and provide the model result with the proposed model. And also provide the visual results of
other compared models.

The authors increased the model accuracy. I have a question about this. While the accuracy is increased, how detection time was changed according the previous models? How many seconds or milliseconds did authors lose or gain with the new model? Could you please compare the training times? Also detection times during test process.

1- Page 3 Line 108
Graph Convolutional Networks][5,25–27] (GCNs) later
Please remove ]
There may be similar writing errors that I could not see. Please check again.

Reviewer 2 Report

Comments and Suggestions for Authors

1, is there a link between those 2 tasks given in fig 1 or they're 2 separate parts?

2. " Our model, unlike most current methodologies, does not employ global average pooling for frame feature to video feature computation.  Instead, we opt for a weighted integration approach, where each frame’s weight is correlated with the textual features, allowing for a semantic-rich video feature conducive to accurate recognition tasks" Why? did you compare the performance of these both approaches to select the weighted approach?

3. ok, we see improvements on table 1, 2 and 3. but what about the computation time? is it really worth to have a 1 or 2 % improvement if the computational time becomes larger and more costly?

4. same comments for table 4. 

5. are actionclip and PoseConv3D only algorithms used in these datasets studied in the literature? 

Reviewer 3 Report

Comments and Suggestions for Authors

The article presents a contribution to the field of human action recognition by introducing the SkeletonCLIP++ model, which integrates techniques for video and text feature analysis. The authors present several key features, including the Weighted Frame Integration (WFI) module, Contrastive Sample Identification (CSI), and the incorporation of the BERT pre-trained model as a text coder. Overall, the article provides a clear description of the methodology and results, making it of interest to CV and NLP researchers.

Positive aspects:

1) The introduction of WFI and CSI. The adoption of a weighted approach to frame processing and the use of the CSI task for precise identification of similar actions can improve recognition accuracy.

2) The use of the pre-trained BERT model with semantic features, improving its ability to interpret textual information.

3) The authors conduct an analysis of experimental results on several corpora such as HMDB-51, UCF-101, and NTU RGB+D 60, demonstrating the applicability and effectiveness of the proposed model in various action recognition tasks.

Negative aspects:

1) Although described in the article, more attention to implementation details and code transparency could be beneficial for other researchers to reproduce results and extend the methodology.

2) The article mainly focuses on the positive results and the effectiveness of the proposed model. However, a more detailed analysis of potential limitations and scenarios where the model might not perform optimally would contribute to a deeper understanding of its applicability.

Conclusion:

SkeletonCLIP++ represents a scientific contribution to the field of human action recognition. Despite the suggested improvements, the authors could consider providing more implementation details and additional analysis of the model's weaknesses. Overall, the article deserves attention and can be recommended for further consideration after the suggested changes have been addressed.

Comments on the Quality of English Language

Moderate editing of English language required.

Round 2

Reviewer 1 Report

Comments and Suggestions for Authors

The paper is ready for publishing. My decision is accept in present form.

Reviewer 2 Report

Comments and Suggestions for Authors

all my questions answered properly. 

Reviewer 3 Report

Comments and Suggestions for Authors

The authors of the article have made the necessary changes, and a repository has been created to reproduce the results. In this form, the article will be useful to the scientific community. Accept in this form.

Comments on the Quality of English Language

Minor editing of English language required.